# Vegetation Cover Dynamics in the High Atlas Mountains of Morocco

Thanh Thi Nguyen [1,†] , Nacer Aderdour [1,2,†] , Hassan Rhinane [2,*] and Andreas Buerkert [1]

1   Organic Plant Production and Agroecosystems Research in the Tropics and Subtropics, University of Kassel, 37213 Witzenhausen, Germany
2   Fundamental and Applied Geosciences, Department of Geology, Faculty of Sciences Aïn Chock, University Hassan II-Casablanca, B.P 5366Maarif, Casablanca 20100, Morocco
*   Correspondence: hassan.rhinane@univh2c.ma
†   These authors contributed equally to this work.

**Abstract:** Since the 1990s, Morocco's agriculture has been characterized by the co-existence and transformation of both modern and traditional smallholder systems. In the Atlas Mountains, the effects of rural–urban transformation have led to intensified irrigated agriculture in some agricultural areas, while others were abandoned. To better understand these effects, this study aimed at (1) analyzing the land use and land cover (LULC) changes, (2) assessing the structure and dynamics of vegetation, and (3) comparing a Support Vector Machine (SVM) classification approach with a seasonal rules-based approach. We, therefore, employed a semi-automatic supervised classification of LULC using Landsat data from the 1990s to the 2020s to distinguish between Open Canopy Vegetation, Bareland, Forest, and Water. Overall accuracies achieved ranged from 88% to 90% in 1990, 2000, 2010, and 2020. SVM results indicated the share of Bareland as >80% of the landscape in all periods. With the seasonal rules-based approach, 10% less Bareland was detected than with the SVM approach. Our findings indicate the limitation of detecting vegetation reflectance in semi-arid mountainous regions such as that prevailing in Morocco using a single machine learning method.

**Keywords:** season effect; remote sensing; land cover change; vegetation

## 1. Introduction

Changes in land use and land cover of rural and connected urban areas are a major consequence of global environmental change [1], which itself is driven by population growth that puts traditional land use, as well as land cover and related ecosystem services, under pressure [2–4]. A central challenge for sustainable development and governance policies at all levels is to effectively manage such transformation processes while maintaining or even enhancing environmental quality, people's livelihood strategies, and food security [5]. In recent years, public awareness has risen for interdependencies between resource use and sustainable development. With the formulation of the 17 UN (United Nations) Sustainable Development Goals (SDGs) and the implementation of these goals into the political agendas of many countries [6], the global commitment towards facing these interdependencies has crystalized. During this process, it also became apparent that research about sustainable development at the regional level remains very fragmented and has a strong "northern" bias, with an underdeveloped assessment and mitigation capacity in poorer tropical and subtropical countries, particularly in North and West Africa [4].

The High Atlas, with its many oases and abundant pastures, provides major ecosystem services as it feeds many rivers in the country, harbors rich plant biodiversity, and preserves traditional agricultural heritage as one of the "cradles" of Moroccan culture. The local forests, comprising the world's biggest area of holm oak (*Quercus ilex* L.) and juniper (*Juniperus phoenicea*, *J. oxycedrus*) at high altitude (up to 2500m ASL) are associated with rainfed agriculture dominated by olive trees (*Olea europaea* L.) and barley (*Hordeum vulgare*

L.), combined with summer pastoralism. The irrigated terraces are limited to low altitudes (about 500 m ASL) along valleys or at higher altitudes (up to 2500 m ASL) along oases with permanent water sources [7]. The period since 2011 experienced prolonged droughts with a precipitation decline by 5 to 30%, which makes the ecosystems in the High Atlas increasingly vulnerable to climate change [8,9]. These droughts also affect oasis agriculture in the Atlas Mountains, which for many centuries have resisted climate irregularities and are presently facing additional challenges from rampant rural–urban transformation. This leads to the emigration of the young generation to Morocco's major cities or even to Europe. Oasis populations suffer from poor educational opportunities, limited infrastructure, and often lack access to countrywide and global value chains for their agricultural products, which makes income generation for sustainable livelihoods difficult. While farmers' agricultural basis rapidly erodes, their economic survival often depends on remittances transferred from relatives migrated to urban areas as well as on governmental subsidies. A more reliable quantification of the LULC changes in the High Atlas Mountains may facilitate the understanding of transformation processes at the oasis level and their effects on livelihood strategies and food security of local farming communities.

During recent decades, remote sensing has become a powerful tool to analyze changes in LULC. Its value is further increased by the recent incorporation of machine learning, such as Support Vector Machine (SVM) and artificial intelligence approaches for image analysis and the classification of features using machine learning. These methods allow automatic classification with higher speeds, but require major sampling and training efforts. The detection and monitoring of vegetation cover, particularly forest cover in arid and semi-arid regions (often referred to as dryland forests), differs from that it tropical forests. Different mixtures of deciduous and evergreen plant populations at a variety of stand densities may lead to an underestimation of the vegetation cover using remote sensing techniques [10–13]. Thus, using vegetation indices with temporal greenness rules was found useful to assess heterogeneous vegetation areas with their own characteristics.

In view of the above, this work aims to analyze the extent of LULC changes of Morocco's High Atlas Mountains. The specific objectives are to (i) quantify major landscape transformation processes between 1990, 2000, 2010, and 2020, (ii) assess the structure and dynamics of vegetation in this region, (iii) compare a SVM classification approach with a seasonal rules-based approach to understand the spatial and temporal distribution of vegetation communities, and (iv) establish cause–effect relationships of changes in ecosystem services in a social-ecological framework.

## 2. Materials and Methods

### 2.1. Study Area

The High Atlas Mountains in northern Morocco stretch across an area of about 55,351 km$^2$ from the country's Atlantic Coast in the West to the Algerian border (Figure 1). They cover an altitude from 598 m to the summit of Jbel Toubkal at 4167 m. The area has a semi-arid climate, with two contrasting seasons: a wet period from October to May and a dry period from June to September with a total annual rainfall from 150 mm in the plain to about 800 mm in the high mountains [14,15]. The High Atlas also provides 85% of the irrigation water for the large plains surrounding the mountains, including Haouz in the north, and Souss, Drâa, and Dades in the south [14]. In 2020, the total population of the High Atlas amounted to 5.2 million, compared with 3.9 million in 1994 (HCP 1994), of which around 90% are Amazigh (Chleuhs) and 10% speak Arabic (Darija). Family incomes are largely based on agriculture and pastoralism [16], whereas tourism plays an increasing role in areas such as the Ourika Valley near Marrakech of nearby Jbel Toubkal.

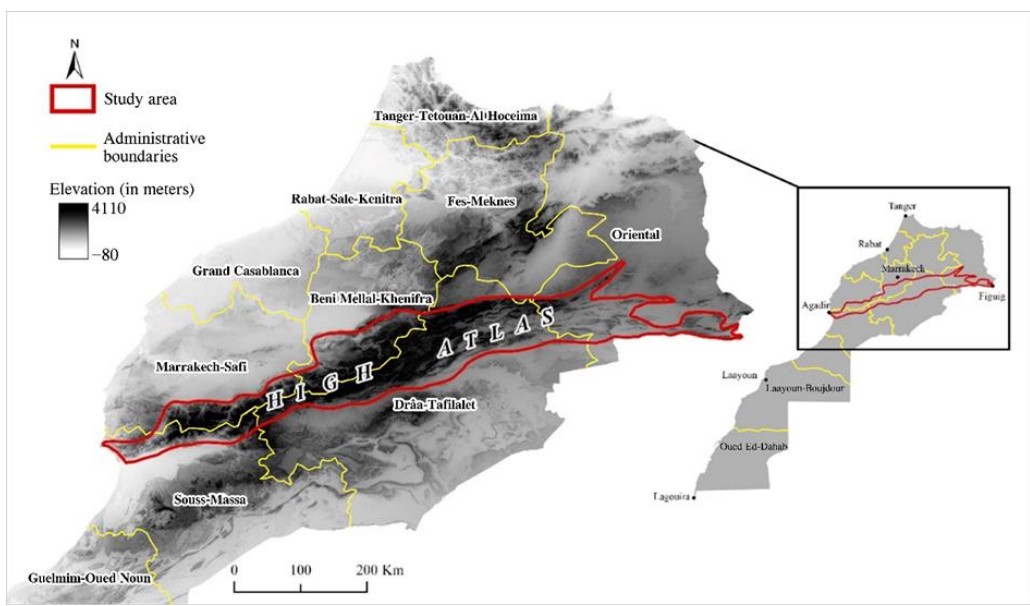

**Figure 1.** Study area, elevation, and boundaries of the High Atlas Mountains in Morocco (Sources: Country boundaries and digital elevation model: "Humanitarian Data Exchange", accessed 20 January 2022, https://data.humdata.org/).

### 2.2. General Framework

In this study we employed an SVM approach, a common and powerful method, to distinguish four vegetation classes, and an evidence-developed method, here named as a "rules-based classification", with four seasons defined by a Normalized Difference Vegetation Index (NDVI; Figure 2). Data analysis comprised a compilation of remote sensing imagery and fieldwork, followed by image processing, a comparison of both approaches, and the harmonization of the classifications.

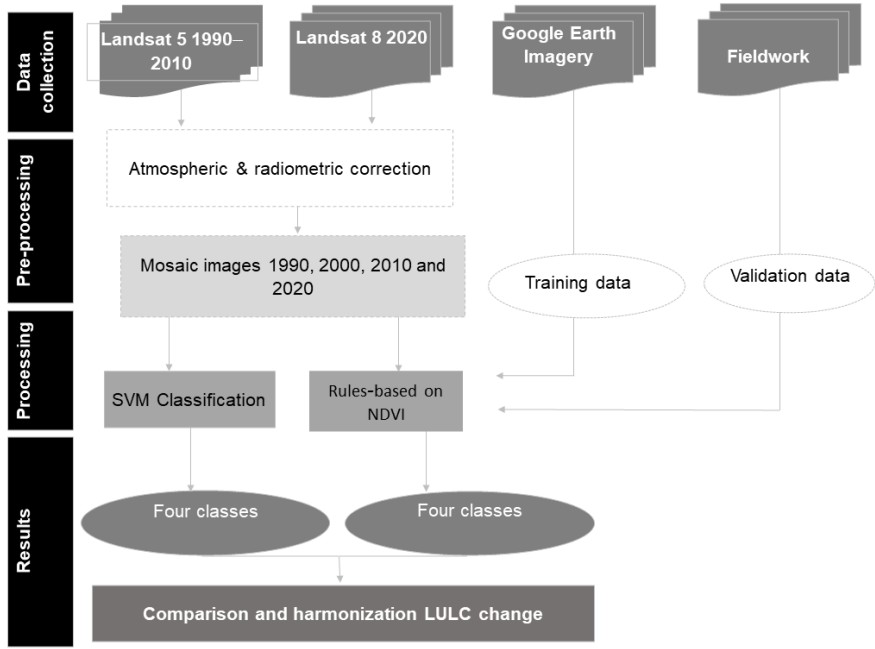

**Figure 2.** Work flow employed in our study of land use land cover changes in the High Atlas Mountains of Morocco.

*2.3. Data Processing and Analysis*

2.3.1. Imagery Acquisition and Pre-Processing

Free Landsat images of four periods with ten years intervals each, 1990, 2000, 2010, and 2020, were downloaded from the United States Geological Survey (USGS, accessed 20 January 2022, "earthexplorer.usgs.gov/"). We used images for the dry season months between July and September with < 10% cloud cover, whereby the time gap between the images needed to be > 10 days. This led to a total of 37 images (Dataset 1, Table 1). To obtain seasonal NDVI within a year, an additional 127 Landsat images (Dataset 2, Table S4) in four periods of each year (1990, 2000, 2010, and 2020) were collected to calculate comparative NDVI values. We omitted Landsat 7 data due to their sensor errors; the correction process showed a large gap in our study area. Instead, we used Landsat Level 1 for time series analyses. This avoided errors of processed Level 2 products that contain two different surface reflectance algorithms.

**Table 1.** Data of Landsat images (Dataset 1) used for classification of land use land cover changes in the High Atlas Mountains of Morocco.

| Image | Sensor | Month/Year | Resolution | Cloud Cover | Nr. of Images |
|---|---|---|---|---|---|
| Landsat 5 | TM | July–September 1990 | 30 m | <10% | 10 |
| Landsat 5 | TM | June–August 2000 | 30 m | <10% | 9 |
| Landsat 5 | TM | June–September 2010 | 30 m | <10% | 10 |
| Landsat 8 | OLI | June–September 2020 | 30 m | <10% | 8 |

The radiance measured by any given system from any object is influenced by factors such as changes in scene illumination, atmospheric conditions, viewing geometry, and instrument response characteristics [17]. Radiometric calibration of satellite-acquired data is therefore essential for quantitative scientific studies, as well as for a variety of image-processing applications [18]. The objective of atmospheric correction is to retrieve the surface reflectance (which characterizes the surface properties) from remotely sensed imagery by removing atmospheric effects.

Using a Dark Object Subtraction (DOS) approach during pre-processing, all images were corrected for atmospheric and radiometric errors. This allowed the removal of haze values caused by scattering of the remote sensing data [19], and the effects of water vapor in the atmosphere, that can absorb the radiation in a specific area [20]. After correction, the images of each year were mosaicked using the Seamless Mosaic tool of ENVI 5.3 (Exelis Visual Information Solutions, Boulder, CO, USA) and clipped to the study area of the High Atlas Mountains.

2.3.2. Classification and Accuracy Assessment

Approach 1. Support Vector Machine

Field dataset: The first of the four vegetation classes, referred to as "Open Canopy Vegetation", represented agricultural lands, grasslands, and tundra. The second class, referred to as "Water", represented artificial lakes, rivers, dams, streams, and reservoirs. The third class, "Forest", included woody and other wild vegetation, while "Bareland" referred to sand dunes, exposed rocks, deserts, and uncultivated areas (Figure 3). Those classes derived from separation [21] after inspecting them using Google Earth 7.1.

The training dataset was collected from the Google Earth Imagery and examined for spectral signatures of each land-use class across images of 2000, 2010, and 2020. We first used a visualization of land cover classes in the most recent year, 2020, employing Google Earth Imagery, and then re-inspected the land cover type for a single point in 2010 and 2010 to build the training set of 2010 and 2000. Moreover, we used false color images of 1990 to inspect land use classes. A small polygon was generated for each class in combination with the false color in the historical imagery of 1990. In total, 482 samples averaging 4000 m$^2$ in size were collected for all periods (Figure 3). Overall, we collected 482 samples, of which

177 referred to Open Canopy Vegetation, 12 to Water, 95 to Forest, and 198 to Bareland (Table 2, Figure S3). During fieldwork conducted in February 2022, we collected 215 GPS points for validation based on Google Earth online images. These points were converted from the Keyhole Markup Language (KML) to shapefile format. On the other hand, we randomly identified a number of points for each class in ArcGIS and exported them in KML format into Google Earth. These points allowed us to verify each class and to conduct an accuracy assessment.

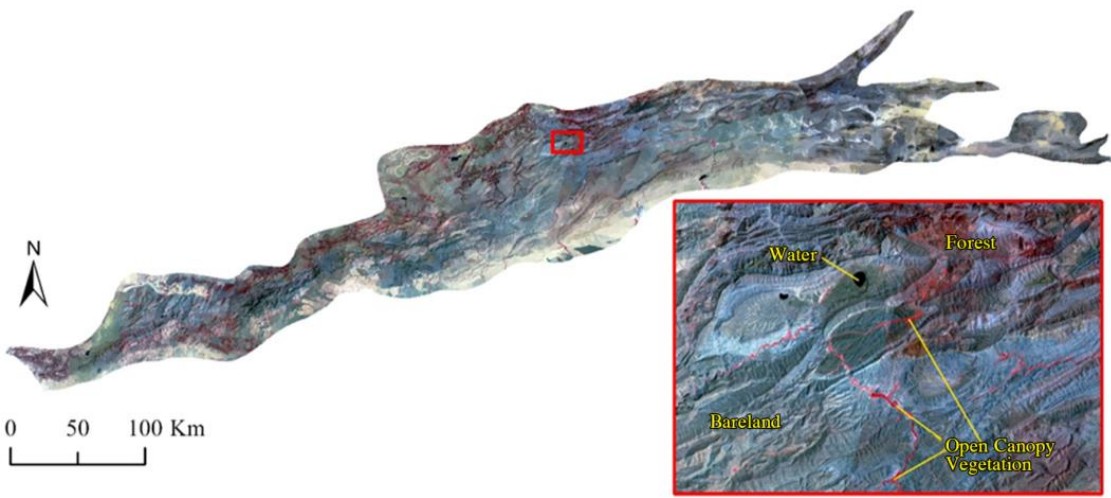

**Figure 3.** False color mosaic of the High Atlas Mountains in 1990 based on Landsat images.

**Table 2.** Type of classes used for the classifications, and detailed descriptions of what each class covers, with the samples collected for the SVM classification of vegetation in the High Atlas Mountains of Morocco.

| Land Cover Class Name | Description | Samples |
|---|---|---|
| Open Canopy Vegetation | Agriculture lands, grass lands, tundra | 177 |
| Water | Artificial lakes, rivers, dams, streams, reservoirs | 12 |
| Forest or Closed Canopy Vegetation | Woody vegetation, wild vegetation with closed canopy and dense tree | 95 |
| Bareland | Sand dunes, exposed rocks, deserts, uncultivated areas | 198 |

Classification and accuracy assessment: The SVM classification was performed through ENVI classic version 5.3 (Exelis Visual Information Solutions, Boulder, CO, USA). This method is considered as a supervised learning system based on statistical learning to identify and distinguish an optimal separation between the classes [21] based on the training [22]. The SVM approach was introduced first in the late 1970s [21]. It is an efficient classifier in high-dimensional spaces, which makes it particularly applicable to multi-dimensional remote sensing data [23]. Classification accuracy was determined by calculating the overall accuracy and the Kappa index [19,24]—a commonly employed index—whereby we used our ground truthing points from the field visit as an independent dataset.

Approach 2. Rules-based using Normalized Difference Vegetation Index

The NDVI was calculated using the following equation for the Normalized Difference Vegetation Index (NDVI, [25]:

$$NDVI = (NIR - RED)/(NIR + RED)$$

where NIR refers to spectra reflectance of the Near Infrared.

NDVI values range from −1 to 1. The highest value (NDVI =1) represents a fully healthy vegetation, while the lowest NDVI value (NDVI = −1) indicates non-vegetative land cover [25].

To investigate the correlation between the NDVI and the land-use types in the High Atlas Mountains, we analyzed Landsat images for the four seasons of each year using the criteria presented in Table 1 [26]. Hereby, Dataset 2 consisted of 127 Landsat images of 1990, 2000, 2010, and 2020 in Period 1 (December–January), Period 2 (March–April), Period 3 (July–August), and Period 4 (Oct–November) within periods < 60 days depending on the availability of historical images (Supplement Tables S1 and S4). All datasets were corrected and mosaicked before calculating NDVI. We randomized 150 points inside the boundaries of the study site area, which we virtually visited using Google Earth Pro 7.1. At each 'virtual ground truthing point', we labeled the class as in the land cover classification of 2020. Vegetation types such as cropping system (mono-cropping, crop and tree), and tree density were determined for the same periods. The NDVI value of each point in the four periods was recorded using the identification function in QGIS. Due to the dominance of Bareland in the High Atlas regions, we collected 88 additional stratified 'virtual ground truthing points' for vegetative areas. To this end, we identified land cover and vegetation type in each of the four periods (four NDVI values of 238 points) to compare the greenness of each land-use/or vegetation type using pairwise comparisons. The distribution and vegetation status allowed us to build rules to distinguish land cover types and to compare them with the results from the first approach using the SVM method. While NDVI represents a consistent index based on the reflectance of greenness in the ground retrieved from remote sensing data, we applied the same rules-based approach built for 2020 to classify land, using the historical NDVI of 1990, 2000 when no field data were available. The Landsat data of 2010 displayed a line error between single images during the mosaicking process; we therefore excluded this period in this approach. We assume this error may be the result of the limited mosaicking capacity in the software used.

To assess the accuracy of our method, we simplified the accuracy assessment by using a qualitative approach with semi-random windows visualization, comparing the classified map versus available high-resolution based maps in QGIS. For this purpose, we randomized seven windows with an average size of $5 \times 5$ km and subsequently visually compared classified maps of the two approaches versus a true color base map. Finally, we employed a harmonization process to group the equivalent class of both approaches. The equivalent land-use type in the SVM approach and the vegetation type (NDVI approach) were grouped into a generic class to compare both approaches using visualization. This allowed the verification of the vegetation differences between both approaches and the vegetation dynamics in the High Atlas Mountains.

## 3. Results and Discussion

### 3.1. Land-Use Classification Using the SVM Method

The results of the classification showed that the High Atlas is largely characterized by Bareland, followed by Forest, Open Canopy Vegetation, and Water. The latter accounted for <0.5% throughout the studied periods. Accuracy assessments obtained through the validation process of the land-use classification and the overall accuracy and Kappa coefficient were satisfactory (Table 3).

**Table 3.** Accuracy assessment in (%) from 1990 to 2020 land-use classification using the SVM method in the High Atlas Mountains of Morocco.

| Land-Use Class | Year 1990 | | Year 2000 | | Year 2010 | | Year 2020 | |
| --- | --- | --- | --- | --- | --- | --- | --- | --- |
| | User | Producer | User | Producer | User | Producer | User | Producer |
| Open Canopy Vegetation | 86% | 100% | 80% | 82% | 100% | 92% | 100% | 92% |
| Water | 85% | 100% | 98% | 100% | 100% | 100% | 100% | 100% |
| Forest | 85% | 87% | 81% | 64% | 97% | 96% | 85% | 85% |
| Bareland | 100% | 82% | 100% | 100% | 84% | 100% | 94% | 89% |
| | Overall accuracy 88%, Kappa coefficient 0.8 | | Overall accuracy 85%, Kappa coefficient 0.7 | | Overall accuracy 90%, Kappa coefficient 0.8 | | Overall accuracy 90%, Kappa coefficient 0.8 | |

Our classification method showed an increase in Forest in 2010 by +1662 km$^2$, which strongly shrunk by −5135 km$^2$ in 2020. During the same period, Bareland was reduced by −1737 km$^2$ followed by an expansion of + 5049 km$^2$ (Table 4, Figure 4). The Open Canopy Vegetation was more than halved from 315 km$^2$ in 1990 to 137 km$^2$ in 2000 and rose again to 217 km$^2$ in 2020. The results indicated only minor changes in Water during 40 years (Table 4).

**Table 4.** Land-use types in the High Atlas Mountains of Morocco classified from Landsat imagery using the SVM method.

| Year | Area (km$^2$) | | | | |
|---|---|---|---|---|---|
| | Open Canopy Vegetation | Water | Forest | Bareland | Total area |
| 1990 | 314 | 59 | 9350 | 45,619 | 55,342 |
| 2000 | 137 | 59 | 9414 | 45,732 | 55,342 |
| 2010 | 186 | 78 | 11,080 | 43,998 | 55,342 |
| 2020 | 281 | 28 | 3278 | 51,755 | 55,342 |

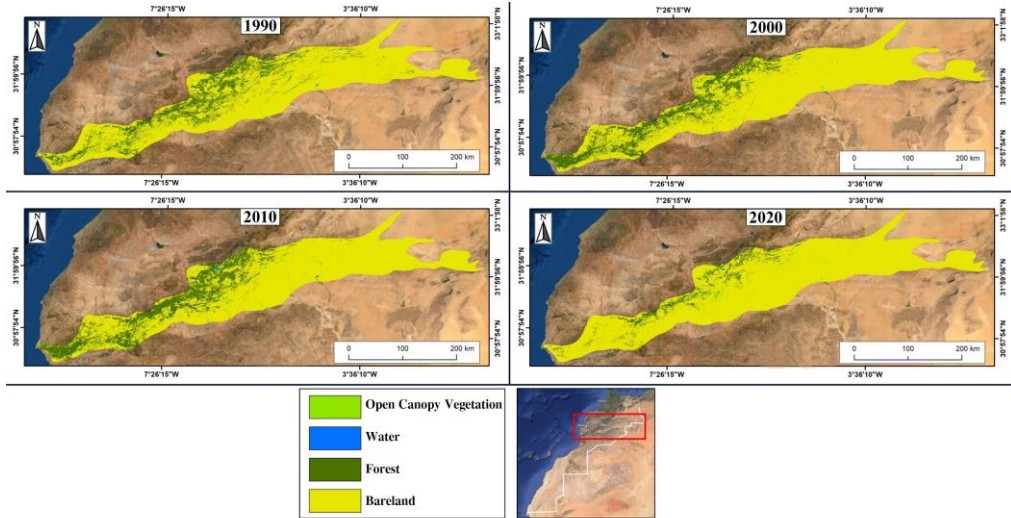

**Figure 4.** Land cover maps of 1990, 2000, 2010, and 2020 derived from Landsat images of the High Atlas Mountains of Morocco using the SVM method.

Detecting changes in forest cover is a challenge in drylands and has received high attention in recent decades [10,11,13]. In these studies, forest degradation was reported largely quantitatively [11,13], and only rarely both quantitatively and qualitatively [10], as we have done. The degradation of forest can be explained by (1) a neo-Malthusian overuse of recourses driven by population increase and concomitant higher demand for firewood and grazing lands, (2) climate change effects such as longer droughts and declining rainfall, (3) consequences of urbanization and the related expansion of intensive agriculture and rangeland, and (4) neoliberal economics promoting agriculture or mining [10,27,28].

Accuracy is a critical aspect of remote sensing image classification, and is typically assessed using User's Accuracy (UA) and Producer's Accuracy (PA). In the case of the forest class, a lower value of UA or PA in the year 2000 compared to other years raises questions about the reliability of the classification results. One possible reason for the lower accuracy in the year 2000 is the similarity in classes, especially in the classification of Forest as Open Canopy Vegetation in some parts of the area. This similarity can lead to confusion in the classification process, resulting in lower accuracy values. It is essential to take this into account when evaluating the results of the classification process.

### 3.2. The Complex Vegetation Landscape Structure in the High Atlas Mountains

Bareland was consistent in color and well distinguishable from other land-use types. The Forest class had a wide range of density due to its scatter and different ages of trees. It comprises (1) dense tree cover (leading to an almost closed canopy), (2) very dense tree cover (closed tree canopy), and (3) coarse tree cover (with scattered tree cover and a widely open canopy). However, using the SVM approach described above, the "Open Canopy Vegetation" class referred to either monocrop stands, mixed crop and tree stands, grass/bushes, and very coarse trees (Figure 5).

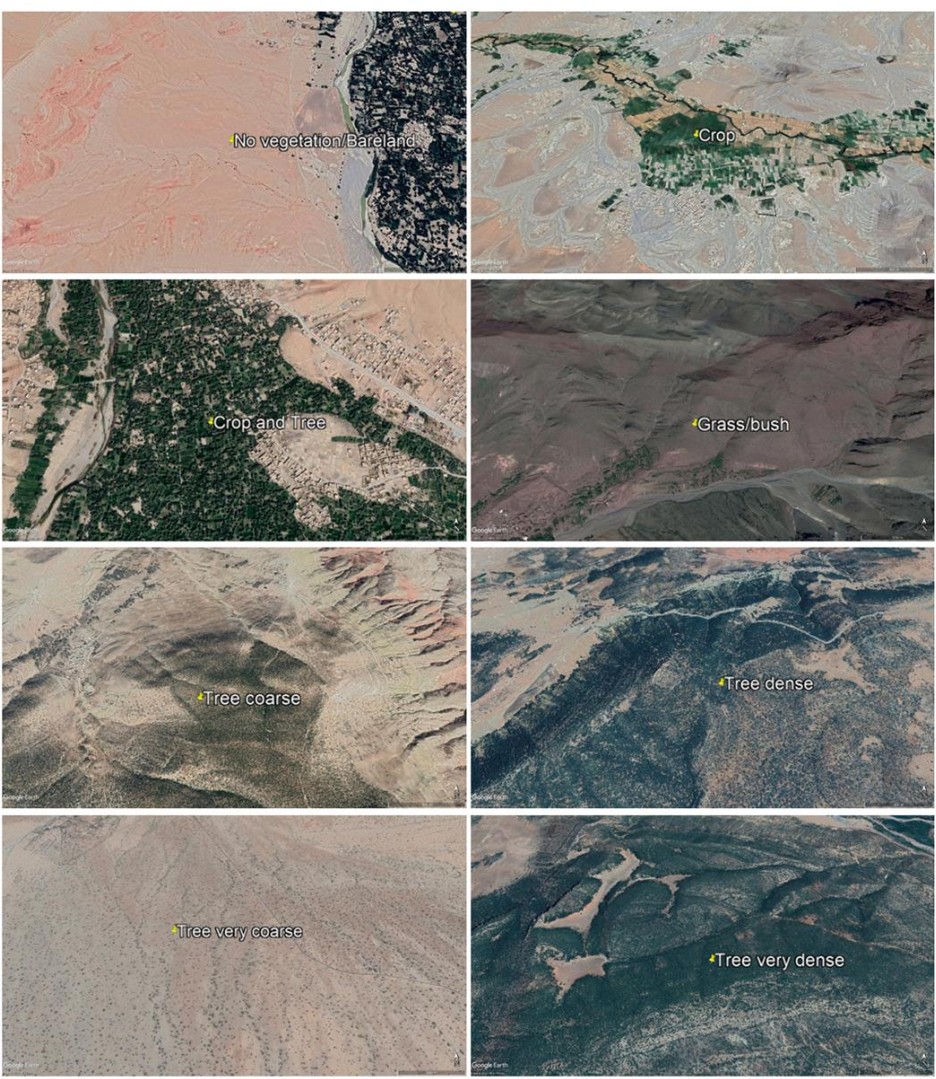

Google Earth Image ® 2022 CNES/Airbus and Maxar Technologies

**Figure 5.** Eight vegetation types classified in the High Atlas Mountains of Morocco. The yellow points represent the location of 'virtual ground truthing points'. All images were downloaded from Google Earth Pro7.1 in 2019 and 2020 at the same scale.

All NDVI values had high deviations within each land cover type and across seasons (Figures 6, 7 and S4). The dominance of values <0.2 through the year indicated large land shares without vegetation cover, even at the end of the rainy seasons in April. The distribution of high NDVI values was quite similar across all four periods, regardless of the season (Figure 6). Statistical analyses (*t*-test) confirmed significantly different NDVI values in April compared with January ($p < 0.001$), August ($p < 0.001$), and October ($p = 0.005$), while NDVI values for the three other periods were statistically not significantly different ($p = 0.324$, 0.323, and 0.844, respectively; Table 5).

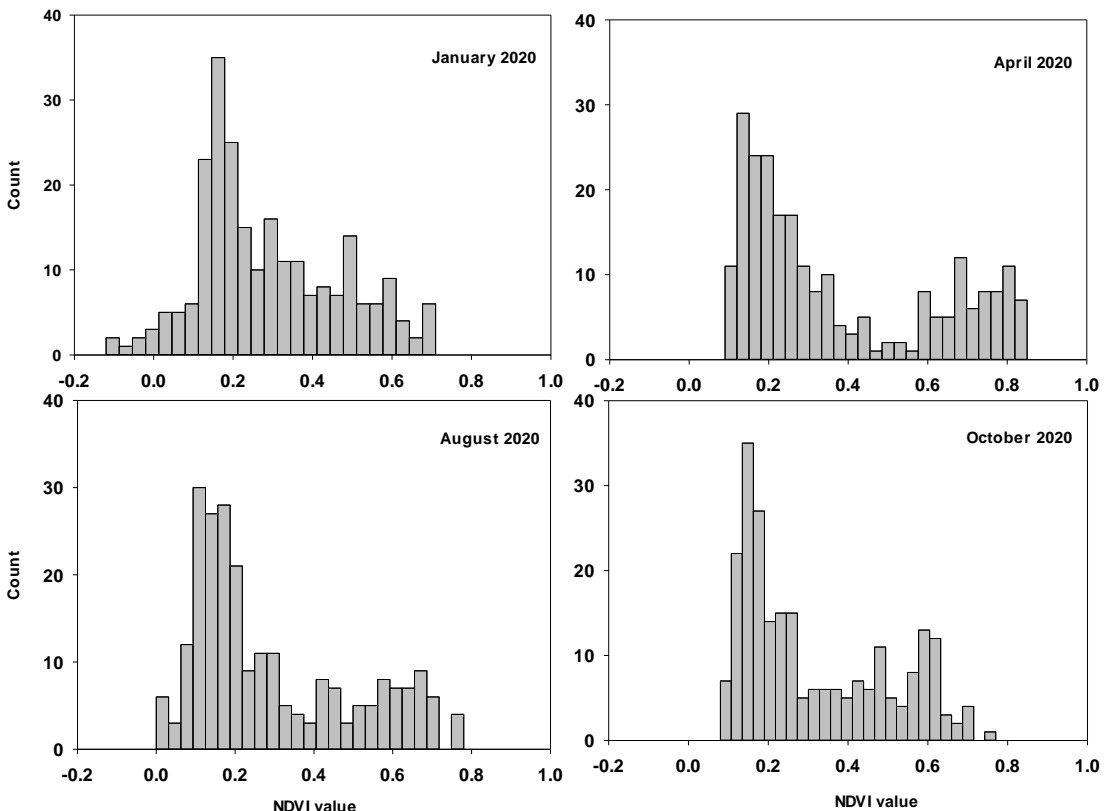

**Figure 6.** Histogram showing distributions of greenness from vegetation for four periods of 2020 in the High Atlas Mountains of Morocco. NDVI was calculated from Landsat 8 (USGS).

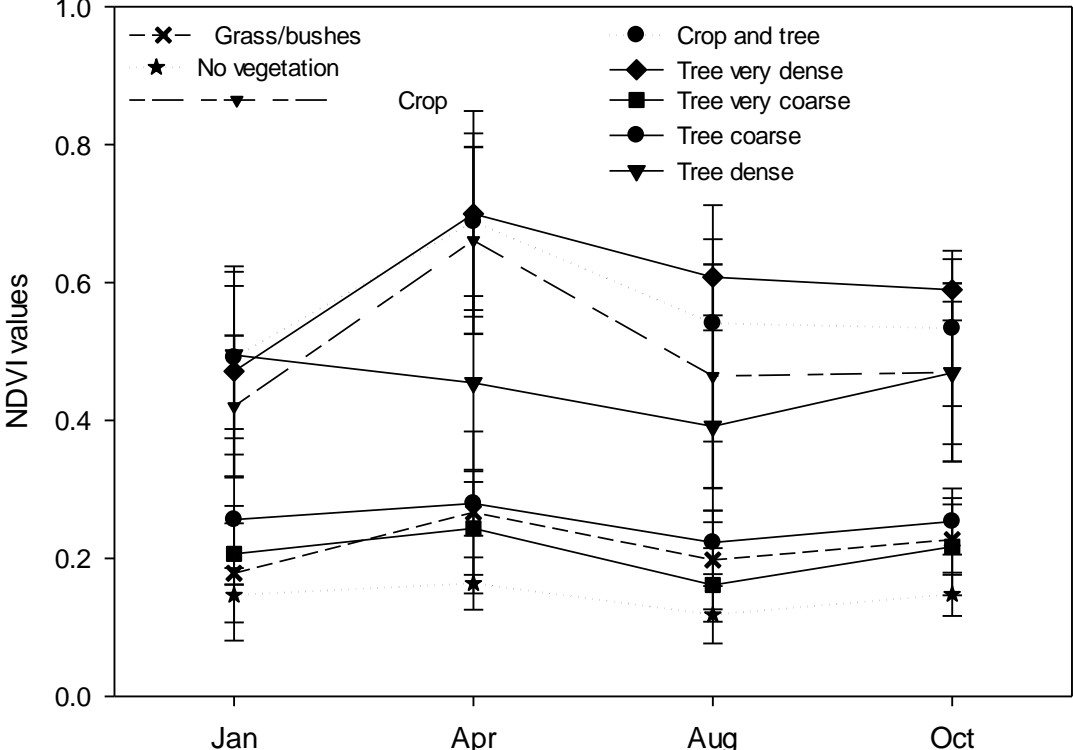

**Figure 7.** Dynamics of NDVI in four periods in the High Atlas Mountains of Morocco, as calculated from Landsat 8 (2020).

**Table 5.** Statistics of NDVI values for the High Atlas Mountains of Morocco in four periods of 2020. The NDVI was calculated from Landsat 8 data. To separate means a pairwise multiple comparison procedure (Holm–Sidak Method) was used (n = 239).

| Comparison | Diff of Means | t | *p*-Values |
|---|---|---|---|
| April vs. January | 0.0889 | 4.812 | <0.001 |
| April vs. August | 0.0853 | 4.615 | <0.001 |
| April vs. October | 0.0603 | 3.265 | 0.005 |
| October vs. January | 0.0286 | 1.547 | 0.324 |
| October vs. August | 0.0249 | 1.35 | 0.323 |
| Augustvs. January | 0.00364 | 0.197 | 0.844 |

High NDVI deviations are in accordance with the findings of [10,29]. They were caused by the diverse forest structure and the complex pattern of cropping and tree plantation in High Atlas Mountains, as described by [30]. The species composition of most forest, composed of oak, cedar (*Cedrus atlantica* (Endl.) Manetti ex Carrière) and sometimes argan (*Argania spinose* (L.) Skeels), was highly heterogeneous, comprising pure stands of deciduous and semi-deciduous trees and mixed stands with evergreen species. Its composition was often hard to determine on Landsat datasets [11]. Moreover, tree density was reported as the main cause of underestimated deforestation from 1970 to 2007 [10]. Changes in forest stands are often reported in terms of area, thereby neglecting changes in density caused by grazing and harvesting of non-timber forest products. This leads to high NDVI dynamics in forest areas, as confirmed by [28] using monthly and annual NDVI data. For agricultural areas, ancient intensive terracing with a combination of different tree species such as almond (*Prunus amygdalus* Batsch), walnut (*Juglans regia* L.), fig (*Ficus carica* L.), and olive (*Olea europaea* L.) at different densities causes highly variable NDVI values within and across seasons. The same applies to irrigated terraces, where the available water not only allows the cultivation of wheat (*Triticum aestivum* L.), barley (*Hordeum vulgare* L.), and recently alfalfa (*Medicago sativa*), but also that of tomato (*Solanum lycopersicum* L.), with maize (*Zea mays* L.), pea (*Pisum sativum* L.), bean (*Phaseolus vulgaris* L.), onion (*Allium cepa* L.), and eggplant (*Solanum melongena* L.) on the same terrace [11]. All of this leads to a wide range of spectrum reflectances in Landsat imagery.

NDVI values were highest for April, whereby monocropped areas, crops mixed with trees, and very dense tree areas had the largest NDVI values during the greenest period, which confirmed the accuracy of our inventory. The SVM approach did not indicate distinct NDVI differences between Forest and Open Canopy Vegetation classes. As for Bareland, the NDVI of grass/bushes, very coarse trees, and coarse tree areas did not allow separation even between dry and rainy seasons (Figure 8).

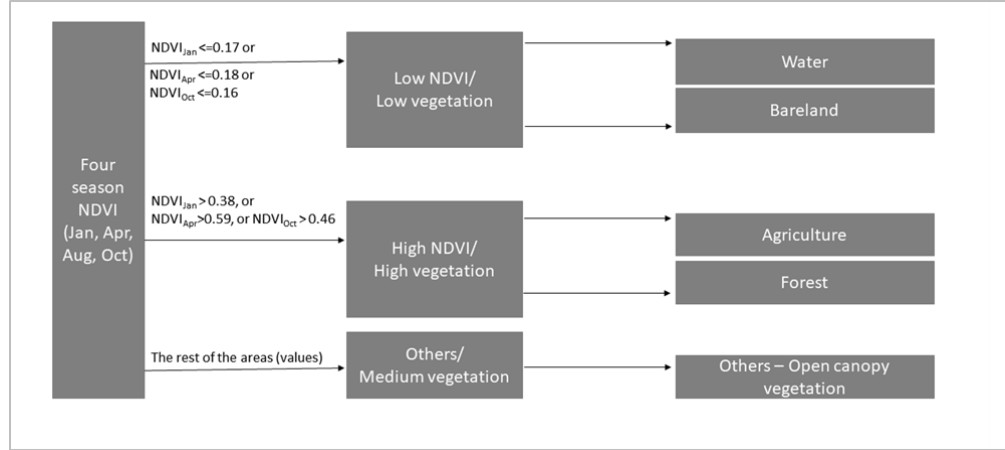

**Figure 8.** Rules-based decision tree to classify land-use and land cover in the High Atlas Mountains of Morocco.

The statistics did not indicate significant spectral differences between closed canopy forest and open canopy vegetation, between Bareland versus Open Canopy Vegetation, or vice versa (Table 6). In particular, the low-density forest even had a numerically lower greenness compared with crops or crops mixed with trees during three of four examination periods (Figure 7).

**Table 6.** Statistics of NDVI values for eight vegetation types (no vegetation, crop, crop and tree, grass/bushes, tree coarse, tree dense, tree very coarse, and tree very dense) in the High Atlas Mountains of Morocco. To separate means a pairwise multiple comparison procedure (Holm–Sidak Method) was used.

| | | | | | | | | **January** |
|---|---|---|---|---|---|---|---|---|
| | No vegetation | Crop | Crop and tree | Grass/bushes | Tree coarse | Tree dense | Tree very coarse | Tree very dense |
| No vegetation | x | <0.001 | <0.001 | <0.001 | <0.001 | <0.001 | 0.054 | <0.001 |
| Crop | <0.001 | x | 0.106 | <0.001 | <0.001 | 0.111 | <0.001 | 0.442 |
| Crop and tree | <0.001 | 0.631 | x | <0.001 | <0.001 | 0.991 | <0.001 | 0.886 |
| Grass/bushes | <0.001 | <0.001 | <0.001 | x | 0.957 | <0.001 | 0.972 | <0.001 |
| Tree coarse | <0.001 | <0.001 | <0.001 | 1 | x | <0.001 | 0.927 | <0.001 |
| Tree dense | <0.001 | <0.001 | <0.001 | <0.001 | <0.001 | x | <0.001 | 0.953 |
| Tree very coarse | <0.001 | <0.001 | <0.001 | 0.632 | 0.551 | <0.001 | x | <0.001 |
| Tree very dense | | 0.578 | 0.895 | <0.001 | <0.001 | <0.001 | <0.001 | x |
| **April** | | | | | | | | |

| | | | | | | | | **August** |
|---|---|---|---|---|---|---|---|---|
| | No vegetation | Crop | Crop and tree | Grass/bushes | Tree coarse | Tree dense | Tree very coarse | Tree very dense |
| No vegetation | x | <0.001 | <0.001 | <0.001 | <0.001 | <0.001 | 0.132 | <0.001 |
| Crop | <0.001 | x | <0.001 | <0.001 | <0.001 | <0.001 | <0.001 | <0.001 |
| Crop and tree | <0.001 | <0.001 | x | <0.001 | <0.001 | <0.001 | <0.001 | 0.087 |
| Grass/bushes | <0.001 | <0.001 | <0.001 | x | 1 | <0.001 | 0.084 | <0.001 |
| Tree coarse | <0.001 | <0.001 | <0.001 | 1 | x | <0.001 | 0.111 | <0.001 |
| Tree dense | <0.001 | 0.999 | <0.001 | <0.001 | <0.001 | x | <0.001 | <0.001 |
| Tree very coarse | <0.001 | <0.001 | <0.001 | 0.261 | 0.331 | <0.001 | x | <0.001 |
| Tree very dense | <0.001 | <0.001 | 0.053 | <0.001 | <0.001 | <0.001 | <0.001 | x |
| **October** | | | | | | | | |

### 3.3. Rules-Based Classification Using NDVI

The SVM solely used the spectral band property, which may lead to misclassification given the vegetation structure of the scattered forests and admixture of trees in agricultural areas of this arid region. Based on the understanding of four seasons NDVI dynamics (Figure 7, Tables 5 and 6), and following a similar approach of Krishnaswamy et al. (2004) [31], we developed a rules-based classification approach. We used similar land-use classes as for the SVM approach, but employed rules-based NDVI slices. The threshold value was defined by the dominant value of 70% of all examined values (Table S2).

When analyzing the variance of the high NDVI group, forest and crop areas could be better distinguished as they showed a significantly different peak value in April ($p = 0.013$; Supplement Table S3). However, the high variance did not allow for further classification; thus, we grouped them into one class equivalent with closed canopy vegetation. Implementing the outcomes from vegetation above, we employed rules-based classification for the most consistent season based on the scatter distribution (Supplement Figure S2) and the relation of greenness to season in order to analyze land-use in the High Atlas as follows (Figure 8):

- Very low NDVI values that represented either Water or Bareland without vegetable cover especially during the dry season: NDVI < 0 represented Water, while ranges of $0 < NDVI_{Jan} <= 0.17$ represented Bareland, equivalent to previous classifications. We used values in January and April that showed more consistent values than other seasons (Figure S2);

- High NDVI values that represented Forest or agriculture areas: $NDVI_{Jan} > 0.38$ and $NDVI_{Apr} > 0.59$ and $NDVI_{Oct} > 0.46$ (Table S2). We excluded values in August, as their distribution had high variances (Figure S2). Values of $NDVI_{Apr} - NDVI_{Aug} > 0.15$ represented agricultural areas. They were assigned as Forest with little differences in NDVI values between the greenest season and the dry season (Table S3);
- Medium value of NDVI represented an Open Canopy Vegetation, which in this case comprised bushes, small and scatter trees except for agricultural areas. This class is denominated as "others".

Despite the strong correlation of NDVI values with precipitation and temperature, they are considered an adequate indicator to detect spatio-temporal changes in land-use (in our case, land cover), especially if they refer to more than one period within a year [28].

### 3.4. Comparison of the Two Classification Approaches

SVM classification techniques often have a high classification accuracy and require only small training data sets while generating satisfying results [32] for heterogeneous areas. In our study, the results of the rules-based vegetation index yielded different areas for each land-use type compared with the SVM approach. Kappa statistics are commonly used to determine the classification accuracy, but their limitations have recently been discussed, especially for mountainous, semi-arid regions. Some authors suggested not using Kappa statistics as the accuracy assessment after Pontius and Millones [33] suggested abandoning the use of this index. However, it is still widely used. Visually, the rules-based vegetation index showed a more compact pattern with the ground rather than SVM in all seven windows. Large areas of forest and agriculture were detected as Bareland according to the results of the SVM approach. Those areas were less green during the dry season (July to September). Forest areas with scattered trees were classified into Bareland, while crop plantations and croplands with mixed trees were identified as Forest (Figure 9).

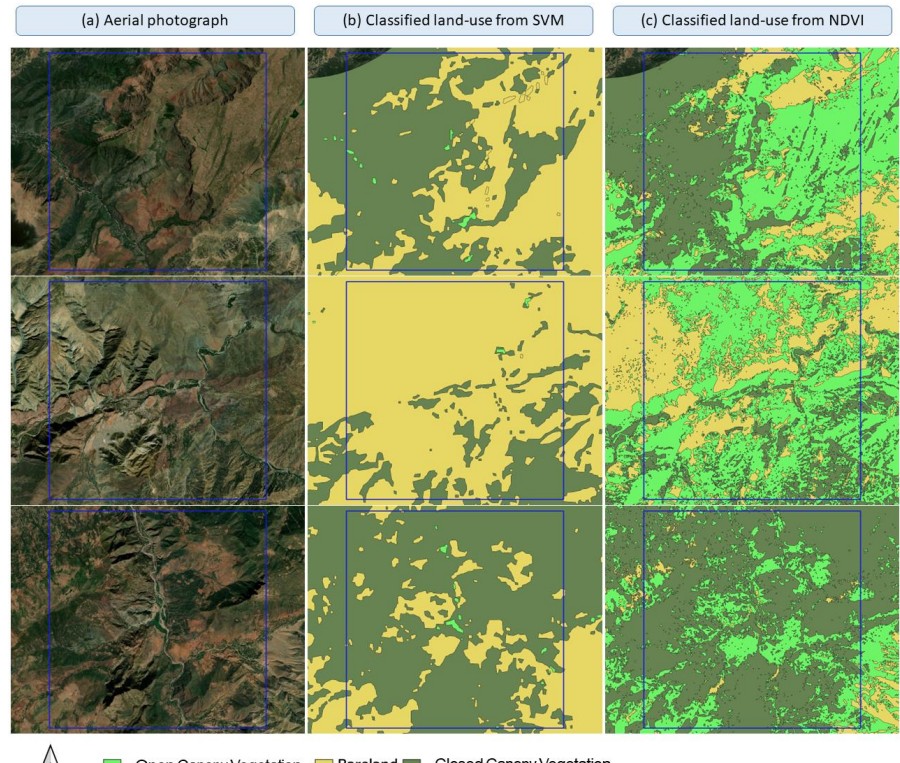

**Figure 9.** Comparison of (**a**) map derived from aerial photograph dataset tile in QGIS, (**b**) land-use classes from the SVM classification approach, (**c**) land-use classes from the NDVI rules-based classification approach in three validation windows (blue) of 5 km × 5 km in the High Atlas Mountains of Morocco.

As the area of Water is very small in the semi-arid study region and the definition of forest and agriculture were inconsistent in both approaches, we harmonized the land-use classes into two classes: Vegetation and Non-vegetation cover (or Bareland). The comparison showed larger vegetation areas using the NDVI rules-based approach than using the SVM approach. Using the NDVI rules-based approach, the Bareland class was assigned a lower proportion than in the SVM approach: 72.0% versus 82.5% in 1990, and 63.3% versus 88.8% in 2020 (Table 7). In 2000, both approaches yielded similar results, which confirmed the high impact of rainfall on vegetation cover [29] as 2000 was a drought year [12] (Figure 10). The rules-based approach considered four vegetation periods rather than including the off-season cropping or vegetation status of deciduous and semi-deciduous trees. Hence, vegetation areas were higher than using the SVM approach [28].

**Table 7.** Difference of vegetation classes in 1990, 2000, and 2020 for two classification approaches of land cover in the High Atlas Mountains of Morocco.

| In 1990 * | NDVI Index Approach | | SVM Approach | |
|---|---|---|---|---|
| | km$^2$ | % | km$^2$ | % |
| Vegetation | 15,482 | 28.0 | 9667 | 17.5 |
| Non-vegetation | 39,904 | 72.0 | 45,684 | 82.5 |
| **In 2000 *** | | | | |
| Vegetation | 7764 | 16.4 | 9551 | 17.3 |
| Non-vegetation | 39,482 | 83.6 | 45,790 | 82.7 |
| **In 2020** | | | | |
| Vegetation | 20,279 | 36.7 | 6212 | 11.2 |
| Non-vegetation | 35,051 | 63.3 | 49,109 | 88.8 |

* Missing 1–2 scenes in 1990 and 2000 caused differences of total area in the comparison; hence, we showed percentages rather than absolute values.

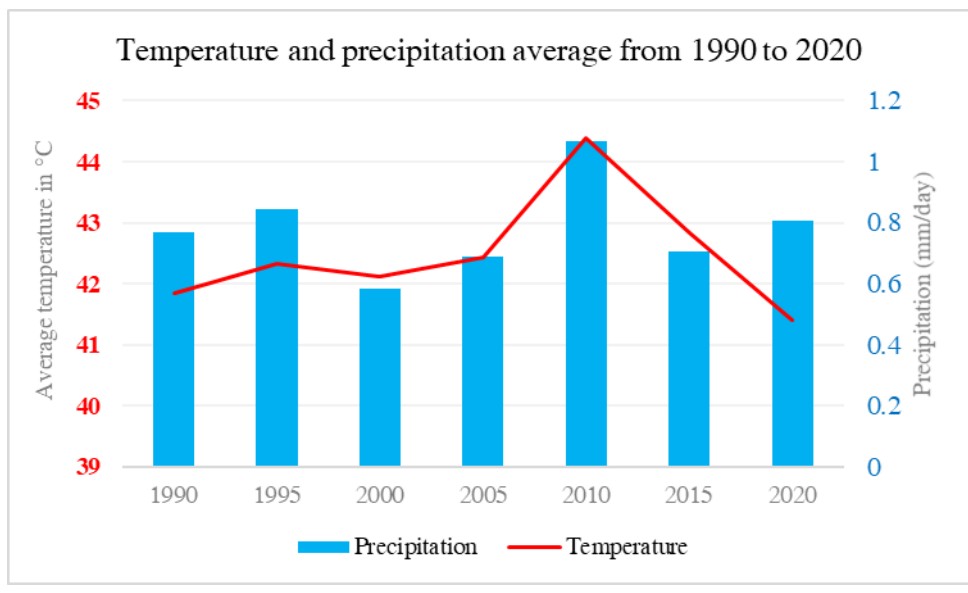

**Figure 10.** Temperature and precipitation average variation from 1990 to 2020 (https://power.larc.nasa.gov, accessed on 21 October 2022).

### 3.5. Combination of Society and Climate Change on Vegetation Cover Dynamic

In Morocco, forest degradation is mainly caused by wildfires and human activity. During 2010–2015, approximately 1% of the forest burned each year, equaling 400,000 hectares, causing, in addition to losses of human lives, enormous ecological and economic damage. Unlike in other parts of the world, where wildfires are of largely natural origin

(mainly lightning), 95% of the fires in Morocco are human-made (criminal, negligence; http://www.eauxetforets.gov.ma, accessed on 15 October 2022). In addition, the removal of firewood and non-timber forest products threaten forest areas and make them vulnerable to dieback. Such regressions of forest ecosystems are particularly strong for cedar stands in the High and Middle Atlas and for cork oak (*Quercus suber* L.) in the Maamora region.

The detected changes in the open vegetation class between 1990 and 2010 are mostly due to variations in precipitation in this area and to urbanization effects. Climate records show a decline in rainfall and a concomitant increase in average temperature between 1990 and 2010 (Figure 10), which is reflected in the vegetation changes indicated by our analyses. Additionally, between 1970 and 2020 many oases in the High Atlas Mountains experienced substantial losses in agricultural land and corresponding increases in fallow and abandoned land. These changes reflect the effects of land abandonment after the emigration of oasis inhabitants, leading to shortages of agricultural labor in irrigated oasis agriculture [7,30].

## 4. Conclusions

Our study demonstrates the importance of understanding the temporal and spatial vegetation dynamics of ecosystems in arid and semi-arid mountain regions. The results show that classification of land cover based on machine learning does not always reflect the dynamics of land cover changes, but rather reflects the effects of multiple factors including those of local climate conditions on vegetation structure. Despite high classification accuracy, as reflected in the Kappa statistics, our findings showed a large misclassification in detecting forest areas. Ground validation using visualization methods to improve such automatic classification is therefore highly necessary. Further research should be conducted to better discriminate between agriculture, forests, and abandoned land, employing a combination of vegetation, water, and moisture indices. This is a prerequisite to investigate land use and land cover transformation processes and their effects on social-ecological systems in the High Atlas Mountains of Morocco.

**Supplementary Materials:** The following supporting information can be downloaded at: https://www.mdpi.com/article/10.3390/rs15051366/s1, Table S1. List of data sources using for SVM method as Dataset 1. Table S2. Threshold of NDVI, sliced from range for low and high NDVI classes at lowest or highest 70% to define the threshold as dominant values. Table S3. T-test between NDVI in April and August, 2020 in High Atlas, Morocco. Table S4. List of Dataset 2 including Landsat 5 and Land 8 (30 m × 30 m) WGS84, UTM 30, downloaded from USGS. Figure S1. Dynamic of land-use classified from Landsat images in High Atlas, Morocco using SVM method. Figure S2. Distribution of low and high NDVI values in four seasons in 2020 in High Atlas Mountainous, Morocco. Figure S3.: Two examples of field data collection in High Atlas in 2020. Figure S4. Examples of NDVI maps in 2020 in four seasons in High Atlas in 2020.

**Author Contributions:** Conceptualization, N.A., T.T.N., and A.B.; methodology, T.T.N. and N.A.; formal analysis, T.T.N. and N.A.; investigation, T.T.N. and N.A.; data curation, N.A and T.T.N.; writing—original draft preparation, T.T.N. and N.A; writing, review and editing, N.A., T.T.N., A.B., and H.R.; supervision, H.R. and A.B. All authors have read and agreed to the published version of the manuscript.

**Funding:** We thankfully acknowledge funding from University of Kassel (Germany) for the first author and from the Erasmus+ Programme of the European Union for the second author. The work greatly benefited from methodological discussions within FOR2432 "Social-Ecological Systems in the Indian Rural-Urban Interface: Functions, Scales, and Dynamics of Transition" funded by the Deutsche Forschungsgemeinschaft (DFG, Project No. 279374797).

**Data Availability Statement:** Not applicable.

**Acknowledgments:** Thanks to Martin Wiehle for his scientific and administrative support. We also thank Kira Fastner for her feedback on NDVI analysis.

**Conflicts of Interest:** The authors declare no conflict of interest.

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
