# Peer review of "Vegetation Cover Dynamics in the High Atlas Mountains of Morocco"

_remotesensing, doi:10.3390/rs15051366_

Round 1
Reviewer 1 Report
The authors present an interesting topic and follow the scope of REMOTE SENSING. Several things still need to be improved, especially citation support and additional discussions to answer research objectives.
In the Introduction, the authors have not conveyed the importance of LULCC analysis in the High Atlas Mountains of Morocco. What benefits will be obtained from the results of the LULCC analysis? What obstacles are often encountered in LULCC research, especially in Morocco?
Line 67-73: The research objectives stated in the abstract are different from those in the introduction section.
L 59-66: there should be citations behind the use of two approaches: the SVM method and a rules-based classification. What are the advantages and disadvantages of the two approaches?
L 105: There should be an explanation why Landsat 7 is not used.
L 134-138: Add the references on how the four groups are classified.
L 142-143: 400 or 482 samples?
L 139-160: This section should mention the citations and equations used to determine the sample number, position, and area size. Likewise, the equation and the minimum number of points are considered in the accuracy assessment. Please note reference [23] does not explicitly discuss overall accuracy and the Kappa index. Please use literature from the primary source.
L197-198: There must be an explanation whether this error was caused by the authors during the processing stage (the authors should have repeated the processing stage), or caused by the Landsat 5 file used.
L 199-206, 384-392: This section should have related references.
L 404: The conclusions do not answer the points of the research objectives.
Author Response
Dear reviewer1,
Attached is our reply to your review

Reviewer 2 Report
Dear Authors,
Your work is well presented and devoted to the highly topical problem in Morocco. In your further research, please, consider and application of soil-adjusted vegetation indices as well as ancillary data to supervised classification (vegetation indices and DEM derivatives, for instance). Usually, it contributes a lot to accuracy enhancement.
Manuscript details:
Journal: Remote Sensing
Manuscript ID: remotesensing-2141493
Type of manuscript: Article
Title: Remote sensing to assess vegetation cover dynamics in the High Atlas
Mountains of Morocco
Authors: Than Thi Nguyen, Nacer Aderdour, Hassan Rhinane *, Andreas Buerkert
Submitted to section: Remote Sensing in Agriculture and Vegetation
In this work, due to the title, the authors describe their experience in applying remote sensing techniques to assess vegetation cover dynamics in the High Atlas Mountains of Morocco. It is devoted to the highly topical problem in Morocco, considering the dramatic decline of forested areas from 2010 to 2020 (more than 300%) discovered in the study (table 4, line 234). The article is well structured, clearly and explicitly written, with plenty of good illustrations. It looks like coming through solid proofreading and makes a good general impression. However, there are some controversial points:
1. Choice of vegetation index. For the rules-based classification, the authors used NDVI, but they didn’t give any reasonable explanation for their choice. It is well-known that NDVI is sensitive to open soils, especially such bright like desert ones. Many experts recommend the usage of soil-adjusted indices in arid and semi-arid landscapes. Despite their statement that “The statistics (of NDVI) did not indicate significant spectral differences between closed canopy Forest and Open canopy vegetation, between Barelands versus Open canopy vegetation, and vice versa” (L295-297), the authors ignored considering other vegetation indices.
2. Validation of the rules-based classification using NDVI. The authors provided an accuracy assessment of the land-use classification using the SVM method (L229-231) but not for the rules-based classification results. In this case, asserting a reliable comparison of the two methods is hardly possible. Moreover, the authors didn’t show the maps derived using the rules-based classification approach. Figure 9 (L361-365) and Table 7 (L377-380) aren’t revealing the classification results by the rules-based approach to the full extent.
3. Atmospheric and radiometric correction. For the classification of land use and land cover changes, the authors used a total of 37 images (L110). To obtain seasonal NDVI within a year, they used an additional 127 Landsat images (L110-111). It should exist a very compelling reason to process all these images manually via the ENVI’s Dark Object Subtraction (DOS) approach (L118) and don’t use Landsat surface reflectance product available via USGS data platforms, including the mentioned “earthexplorer.usgs.gov/”. However, from the text, it doesn’t clear.
4. Land cove & land use. It looks like the authors don’t distinguish these terms and use them as synonyms. In the introduction, the authors write: This work aims at analyzing the extent of LULCC of Morocco’s High Atlas Mountains (L67-68). However, they write only about land-use classification and land use types in the Result and discussion section. In table 2 (L161-163), the authors describe land-uses classes: Open canopy vegetation, Water, Forest or closed canopy vegetation, and Barelands. But due to most of the well-known definitions, these classes refer to land cover, not land use. After all, the authors conclude that The results show that land cover classifications based on machine learning does not always reflect …(L406-407)
5. Poor review of existing approaches to remote sensing applications in LULCC analysis in arid and semi-arid areas.
Several details also have to be revised:
L16: if classification is supervised and include learning/rules setting by an expert, this classification is a semi automatic, not automatic
L52: addition challenges
L95: Why NDVI is specific?
L125-131 The paragraph describes the general knowledge and obviously should be well-known to the readers of MDPI remote sensing
L194-195 Assuming that NDVI represents a consistent index based on the reflectance of light from remote sensing data… Confusing statement and should be reformulated. Light reflects form land cover and recorded by a sensor.
L401: Reference?
L403: Source?
L529: Need to revise in SigmaPlot. Indeed! Please, sign the axes
Author Response
Dear reviewer,
Kindly, attached is the reply to your comments

Reviewer 3 Report
Dear Authors,
I have reviewed the manuscript you submitted to the Remote Sensing (MDPI) Journal, entitled "Remote sensing to assess vegetation cover dynamics in the High Atlas Mountains of Morocco".
Based on my review, my opinion is overall positive, as I think that the paper conveys interesting findings and methods, that are worth a publication in the targetted journal. However, I do have mixed feelings as I have had the impresison that the document/manuscript submitted is still a draft, in many ways: many sources are missing, some comments (not intended to the reader) are still left out in the document, some figures are not properly ordered or formatted, and many other flaws I have been able to detect. As such, I deeply suggest the authors to carefully proofread the manuscript and ensure that the body text and figures and tables are clean before considering re-submitting it. Also, I have added many comments (questions, suggestions, remarks for further discussion) in the manuscript, and I am expecting having point by point answers to the comments.
I suggest Major Revision.

Author Response
Dear reviewer,
Attached is the reply to your review

Reviewer 4 Report
Hello there,
I appreciate your effort on this research. However, I would suggest acquiring a very specific classification schema first for the SVM approach and merging them later to obtain current broad scheme. It will provide you the base for comparing two approaches.
Looking forward to see the improved version,
Good luck!

Author Response
Dear reviewer,
Kindly find in attached the reply to your comments

Round 2
Reviewer 1 Report
The authors have made improvements as suggested.
Reviewer 3 Report
Congratulations on being able to provide timely a thoroughly revised version of the manuscript. I have checked the changes and the responses provided to my concerns, which appear to be very complete. I have no further issue to raise.